# Vortex states in a PbTiO$_3$ ferroelectric cylinder

**Svitlana Kondovych$^{1,2\star}$, Maksim Pavlenko$^3$, Yurii Tikhonov$^3$,
Anna Razumnaya$^4$ and Igor Lukyanchuk$^{3,5}$**

**1** Institute for Theoretical Solid State Physics, IFW Dresden, 01069 Dresden, Germany
**2** Life Chemicals Inc., 02660 Kyiv, Ukraine
**3** Laboratory of Condensed Matter Physics, University of Picardie, 80039 Amiens, France
**4** Jožef Stefan Institute, 1000 Ljubljana, Slovenia
**5** Landau Institute for Theoretical Physics, 142432 Chernogolovka, Russia

$\star$ s.kondovych@ifw-dresden.de

## Abstract

The past decade's discovery of topological excitations in nanoscale ferroelectrics has turned the prevailing view that the polar ground state in these materials is uniform. However, the systematic understanding of the topological polar structures in ferroelectrics is still on track. Here we study stable vortex-like textures of polarization in the nanocylinders of ferroelectric PbTiO$_3$, arising due to the competition of the elastic and electrostatic interactions. Using the phase-field numerical modeling and analytical calculations, we show that the orientation of the vortex core with respect to the cylinder axis is tuned by the geometrical parameters and temperature of the system.

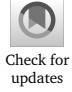

# 1   Introduction

The seminal discovery of vortex matter in nanostructured ferroelectrics has crucially changed our understanding of polarization behavior in these materials [1–3], ranking ferroelectrics among the family of noble topological materials, such as superconductors, magnets, and topological insulators. However, the intrinsic mechanism of vortex polarization swirling in ferroelectrics is unique and originates from the peculiar harnessing of the electrostatics and confinement effects. More specifically, it is based on the tendency of the polarization of the system, $\mathbf{P}$, to avoid the creation of the depolarization charges, $\rho = -\mathrm{div}\,\mathbf{P}$, which induce the energetically unfavorable depolarization electric field [4]. Hence, the polarization structuring with $\mathrm{div}\,\mathbf{P} = 0$ is the principal topological constraint in ferroelectrics [5,6]. Vortices are the representative example of such formations, most commonly observed in ferroelectrics [7–12].

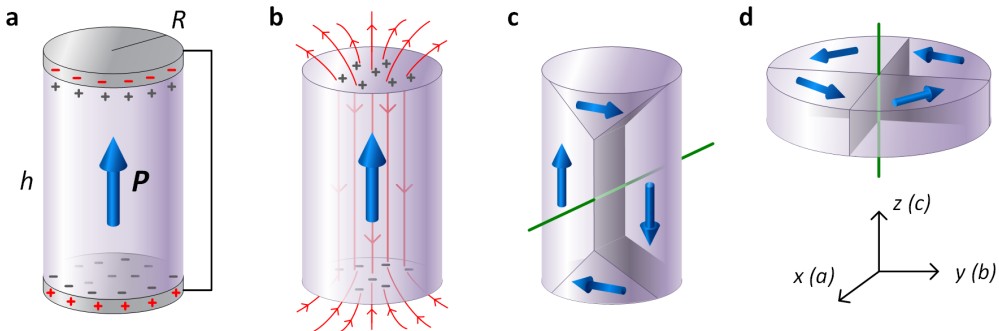

Figure 1: **Formation of vortex states in a ferroelectric cylinder.** (a) A uniform $c$-phase with polarization $\mathbf{P}$ (blue arrow) is stable in a cylinder of height $h$ and radius $R$ with short-circuit electroded edges. The surface depolarization charges (gray symbols) are screened by the electrode's free charges (red symbols). (b) In a free-standing ferroelectric cylinder without electrodes, the depolarization charges at the edges produce the depolarization electric field (red arrows) that finally destabilizes the uniform polarization distribution. (c) An elongated cylinder hosts the stable $a$-vortex state with the core line (shown in green) oriented along the $a$- (or $b$-) axis. This state is alternatively visualized as a polarization flux-closure domain structure with 180° and 90° domain walls. (d) In a disc-shaped cylinder, the vortex core has the $c$-axis orientation. The polarization distribution can be also seen as four domains with polarization flux closure, separated by the 90° domain walls. The orientation of axes is shown at the bottom right corner.

The objective of this paper is to give an insight into the emergence of vortices in nanocylinders of ferroelectric oxides with cubic crystal symmetry. We study lead titanate perovskite, PbTiO₃ (PTO) as an exemplary system. The basic idea of the polarization vortices formation

is sketched in Fig. 1. The uniformly polarized along the cylinder axis state (*c*-phase) occurs in a cylinder with short-circuited edges, see Fig. 1a. The depolarization charges created in the points of polarization termination at the cylinder edges are screened by metallic free charges of the electrodes, hence no depolarization effects arise. However, when the electrodes are removed (Fig. 1b), the unscreened depolarization charges generate the depolarization field, which affects the polarization distribution. Then, the polarization swirls into divergenceless vortex states of either *a*- (*b*-) (for elongated cylinders) or *c*- (for disc-shaped cylinders) orientation of the vortex axis, shown in Fig. 1c and Fig. 1d respectively. Alternatively, these states can be viewed as flux-closure domain structures with 180° and 90° domain walls (DWs).

The basic vortex states were discovered in several cylindrical ferroelectrics. The *c*-vortex states in cylinders, first suggested in [9] as competitive divergenceless states, were studied in the ferroelectric $BaTiO_3$ prolate cylinders, constrained by the non-ferroelectric polymer matrix [13] and also in presence of the flexoelectric effect [14] using the phase-field approach. They were also simulated in strained $PbZr_{0.5}Ti_{0.5}O_3$ nanodiscs [15] and in PTO nanocylinders [16] using the atomistic approach. Here we give the detailed study of different vortex phases in free-standing nanocylinders of PTO and reconstruct the phase diagram of the system as a function of the geometrical parameters of cylinders, their radius, $R$ and height, $h$, and of the temperature, $T$. We reveal the role and delicate balance of the elastic and electrostatic energy contributions in the vortex phase formation and demonstrate that a rich variety of vortex structures is possible within each type of the vortex axis orientation.

Although the emergence of polarization topological structures was cogently demonstrated in nanostructured ferroelectrics [1–3], and the technology nodes of modern ferroelectric-based nanoelectronics are scaled down to the appropriate size of tens of nanometers [17], exploration of topological excitations in ferroelectrics remains an appealing task. Our findings establish a platform for tailoring the vortex structures in nanocylinders to use them as essential components of nanodevices. The discovered multitude of the vortex (meta)stable states allows for the implementation of ferroelectric multi-valued logic [18–21] and neuromorphic elements [22]. Another impact is related to the design of polarization textures in the long ferroelectric nanotubes, nanorods, and nanowires with well-established fabrication technology [23–25]. The obtained nanostructuring of the polarization and depolarization fields in the ferroelectric nanowires can be used for engineering the bias-free tips in the AFM/PFM techniques.

## 2   Model

We model the free energy density functional in terms of the polarization components $P_i$, elastic strains $u_{ij}$, and electrostatic potential $\varphi$ as:

$$\mathcal{F}^{\mathrm{u}} = \left[ a_i(T)P_i^2 + a_{ij}^{\mathrm{u}}P_i^2P_j^2 + a_{ijk}P_i^2P_j^2P_k^2 \right]_{i \leq j \leq k} + \frac{1}{2}G_{ijkl}(\partial_i P_j)(\partial_k P_l)$$

$$-q_{ijkl}u_{ij}P_kP_l + \frac{1}{2}c_{ijkl}u_{ij}u_{kl} + (\partial_i\varphi)P_i - \frac{1}{2}\varepsilon_0\varepsilon_b(\nabla\varphi)^2 , \tag{1}$$

where the sum is taken over the circularly permutated indices $\{i, j, k, l\} = \{1, 2, 3\}$. The first term in square brackets stands for the Ginzburg-Landau (GL) energy written as in [19], where $a_1 = \alpha_1(T - T_c)$, $T_c$ is the transition temperature to the ferroelectric state, and the 4th-order zero-strain coefficients $a_{ij}^{\mathrm{u}}$ are calculated by the Legendre transformation from the stress-free coefficients $a_{ij}^{\sigma}$ [26], as described in [27]. The second term in (1) with coefficients $G_{ijkl}$ corresponds to the gradient energy. The last terms represent the elastic and electrostatic energies, with $c_{ijkl}$ being the elastic stiffness tensor and $q_{ijkl}$ the electrostrictive tensor;

$\varepsilon_0 = 8.85 \times 10^{-12}\,\mathrm{C}\,\mathrm{V}^{-1}\mathrm{m}^{-1}$ is the vacuum permittivity and $\varepsilon_b \simeq 10$ is the background dielectric constant of the non-polar ions. The numerical values of coefficients for PTO, used in calculations, are given in Appendix A.1.

For analytical calculations, we use a simplified isotropic version of (1) without the $6^{\text{th}}$-order terms,

$$\mathcal{F}_{iso}^{\mathrm{u}} = a_1(T)P^2 + \bar{a}_{11}^{\mathrm{u}}P^4 + \bar{G}\mathrm{rot}^2\mathbf{P} - \bar{q}_{12}u_{ii}P_jP_j - \bar{q}_{44}u_{ij}P_iP_j + \frac{1}{2}\bar{c}_{12}u_{ii}^2 + \bar{c}_{44}u_{ij}^2, \qquad (2)$$

in which the averaged coefficients (denoted with the overbar) are selected to match the principal bulk properties of PTO and calculated by the tensor averaging of functional (1), see Appendix A.2 for details. We consider only the depolarization charge-free configurations with $\mathrm{div}\,\mathbf{P} = 0$, resulting in the vanishing electrostatic contribution to the functional (2). Then, the polarization is scaled in units of the uniform polarization of the stress-free bulk sample, $P_0 = \left(|a_1|/2\bar{a}_{11}^{\sigma}\right)^{1/2} \simeq 0.7\,\mathrm{C}\,\mathrm{m}^{-2}$, and the characteristic length scale is defined by the coherence length, $\xi_0 = \left(\bar{G}/\alpha_1 T_c\right)^{1/2} \simeq 1\,\mathrm{nm}$.

## 3 Phase diagram

The resulting phase diagram of vortex states in cylinders of different heights, $h$, and radii, $R$, is shown in Fig. 2. The exemplary polarization textures of these states are sketched on the left and on the bottom of the diagram. As it was mentioned in the Introduction, the competition occurs between the $c$-oriented uniformly-polarized state and vortex states of the different, $a$-or $c$-, orientation of the core. The analytically estimated separation line between these two states is shown in blue in the phase diagram in Fig. 2 and discussed in Section 4.

In the uniformly polarized state, the topological requirement $\mathrm{div}\,\mathbf{P} = 0$ cannot be satisfied. The depolarization charges necessarily arise at the surface of the sample, in particular at the cylinder edges, where the polarization field terminates. The corresponding depolarization fields distort the uniformity of polarization at the edges, hence such state, depicted as state C in Fig. 2, occurs only in very long cylinders with $h \gtrsim 500\,\mathrm{nm}$. The $c$-phase region in such a nanowire becomes shorter as the radius increases. However, further investigation of bigger cylinders does not reveal any new states while being computationally demanding.

The depolarization field is zero or vanishingly small in the divergenceless $a$- and $c$-oriented vortex states with flux-closure polarization texture that emerge in a vast range of cylinder geometries. The realization of the particular state depends on the delicate energy balance of the ferroelectric and elastic contributions, provided by the functional (1). States I, II, and III in Fig. 2, which arise in elongated cylinders with $h \gtrsim R$, correspond to the $a$-vortex states with vertically-stretched, twisted, and horizontally-stretched vortex core, respectively. States IV, V, and VI, which appear in disc-shaped cylinders with $h < R$, correspond to the $c$-vortex states with twisted, non-deformed, and deformed vortex core. The multivortex state VII contains both $c$- and $a$-type vortices.

In the next section, we proceed with the discussion of the stability conditions of different topological states in PbTiO$_3$ cylinders on a quantitative level.

## 4 Discussion

### 4.1 Uniform $c$-phase

We consider first the formation of the uniform state ($c$-phase) in a short-circuited cylinder with the polarization vector directed along the cylinder axis, see Fig. 1a. In this case, the surface

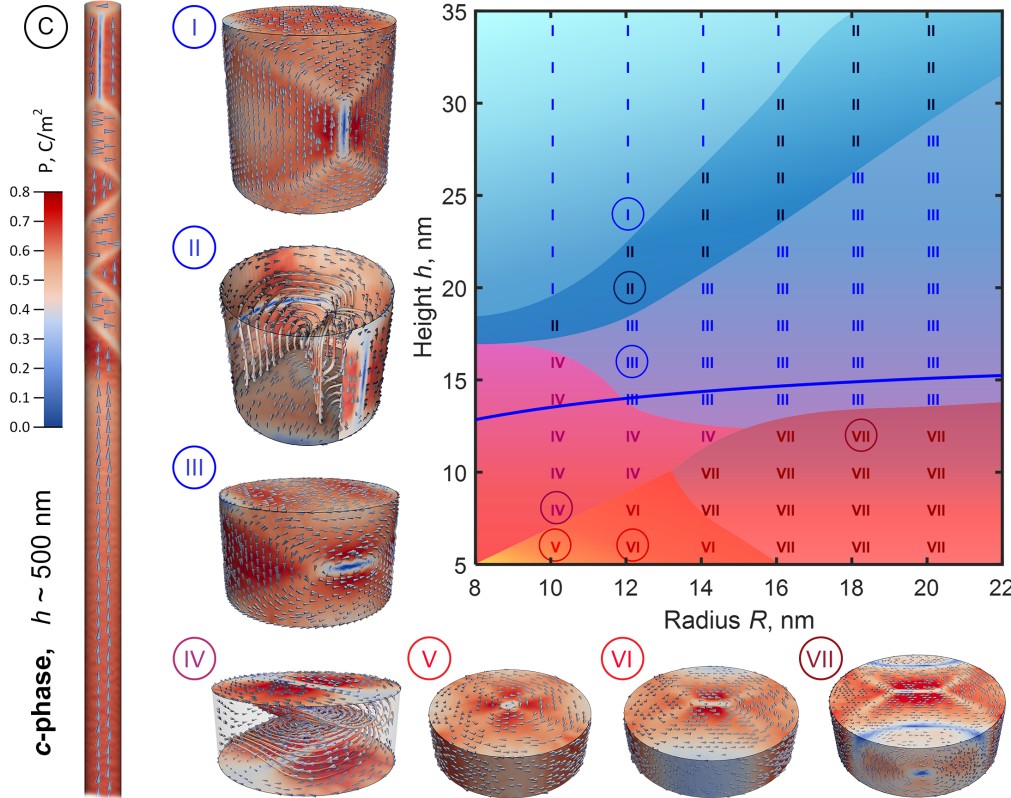

Figure 2: **Vortex states and phase diagram of a ferroelectric cylinder.** State C corresponds to the uniform *c*-phase, occurring in the middle part of very long cylinders with $h \gtrsim 500$ nm and terminated by the nonuniform polarization texture at the edges (only the top part is shown). States I-III are the *a*-vortex states and IV-VI are the *c*-vortex states. The multivortex state VII hosts both *c*- and *a*-type vortices. The orientation and magnitude of polarization are visualized by arrows and colour scale and also, for twisted states II and IV, by the internal streamlines. The phase diagram depicts the stability regions of vortex states I-VII (highlighted by different colours) in *R*-*h* coordinates. The roman numbers indicate the type of the stable states. The circled numbers correspond to the exemplary states shown around the phase diagram. The solid blue curve shows the separation line between the *c*- and *a*-vortex regions calculated theoretically based on vortex energies.

depolarization charges are screened by the electrode charges. Note that, in general, electrodes may significantly affect the polarization distribution. For instance, oxide electrodes with much larger screening length lead to the incomplete screening of depolarization fields and influence the domain structure [28]. However, in this study we restrict our discussion to the ideal case of metallic electrodes. Thus, the energy of the system, $E_0 = \pi R^2 h F_0$, contains only the GL energy density of the stress-free uniform state, $F_0 \approx -0.075 \times 10^9$ Jm$^{-3}$, calculated from (1).

In a free-standing cylinder without electrodes, the total energy,

$$E_{\mathrm{u}} = \pi R^2 h F_0 + 2 E_{\text{œ}}, \tag{3}$$

includes also the terminal energy of two edges, $2E_{\text{œ}}$, arising due to the emergent depolarization field. As discussed in the previous section, it modifies the polarization distribution at the cylinder edges and forms the state of type C (Fig. 2), in which the multiple domains corresponding to *a*- or *c*-vortices appear on approaching the cylinder edges. The huge value of $E_{\text{œ}}$

can be surmounted by the negative energy of the uniform ferroelectric state, remaining in the middle region of the samples only for very long cylinders with $h \gtrsim 500$ nm. Interestingly, the observed perturbed polarization texture at cylinder edges possesses the spontaneously broken chirality, the effect that was thought to occur due to the flexoelectric contribution [14].

Further, we consider the vortex states with $a$- (or $b$-) and with $c$- oriented axes that form to reduce the depolarization energy in the cylinders of moderate and short heights.

## 4.2 $a$-vortices

The $a$-vortex state, arising in an elongated cylinder with $h \gg R$ (Fig. 1c), is significantly deformed. In the bulk of the cylinder, the polarization texture forms a vertical 180° DW of thickness $\sim \xi_0$, having alternative "up" and "down" orientations on both sides of the wall. The DW, hence the vortex core aligns with $a$- or $b$- crystal axis, which are energetically equivalent (for definiteness, we refer to $a$-vortex state). The creation of the DW requires additional energy that scales proportionally to the DW area as $Rh$ and can be estimated as $\xi_0 Rh\, F_{1a}$. The factor $F_{1a} > 0$ stands for the effective energy density stored in the vertical DW and also accounts for the numerical coefficients related to the DW geometry.

Close to the cylinder edges, the $a$-vortex polarization lines make the U-turn with the formation of two horizontal $b$-oriented domains at the top and bottom of the cylinder. These domains are separated from the vertical $c$-domains by the 90° DWs, departing from the termination of the 180° DW. The effective area of these DWs is proportional to the cylinder cross-section area, $\pi R^2$, and their thickness is proportional to $\xi_0$. Hence, the terminal energy acquires the form $E_æ \simeq \pi R^2 \xi_0\, F_æ$, where factor $F_æ$ describes the effective density of the energy stored in the terminal region. Normally, $E_æ$ is smaller than $E_œ$ because the former does not contain any significant depolarization electrostatic contribution.

We estimate the total energy of the $a$-vortex in the elongated cylinder as:

$$E_a = \pi R^2 h\, F_a + \xi_0 Rh\, F_{1a} + 2\pi R^2 \xi_0\, F_æ \,, \tag{4}$$

where $F_a < 0$ is the energy density within the bulk of $c$- and $b$-domains. It is slightly smaller than $|F_0|$ in its absolute value because of the residual long-range elastic contribution produced by the polarization inhomogeneities in the DWs and the vortex core. The second and the third terms correspond to the DW and terminal energies, as described above. The fit of the phase-field simulation results for PTO cylinders (see Appendix A.3) gives the following numerical values of the parameters in Eq. (4): $F_a \approx -0.07 \times 10^9$ Jm$^{-3}$, $F_{1a} \approx 0.50 \times 10^9$ Jm$^{-3}$, and $F_æ \approx 0.29 \times 10^9$ Jm$^{-3}$.

The solid blue line in Fig. 3a shows the numerical solution for the polarization $z$-component distribution in perpendicular to the DW direction for the cylinder with $R = 10$ nm and $h = 50$ nm. The dotted blue line presents the DW profile $P(r) = P_0 \tanh\left(r/\sqrt{2}\xi_0\right)$ obtained analytically within the isotropic model (2), which matches well the numerical result.

When cylinders become shorter, approaching $h \sim 2R$, the vortex structure appears to be more distinct. In the cylinder with $R = 10$ nm and $h = 22$ nm, the polarization demonstrates the non-monotonic dependence (solid red line in Fig. 3a), different from that for the flat DW. On further cylinder shortening, at heights $h$ comparable to $R$, the elastic stretching of the $a$-vortex core changes its orientation from vertical to a horizontal one, passing through the intermediate state with a twisted DW, see states I, II, and III in Fig. 2. Finally, at even smaller $h$, the vortex core changes its orientation from $a$- to $c$-direction.

## 4.3 $c$-vortices

The $c$-vortex states IV, V, and VI (Fig. 2), which form in disc-shaped cylinders with $h \lesssim R$, have different structures. We consider first the vortex state V, arising at small $R \simeq 6$-$10$ nm and

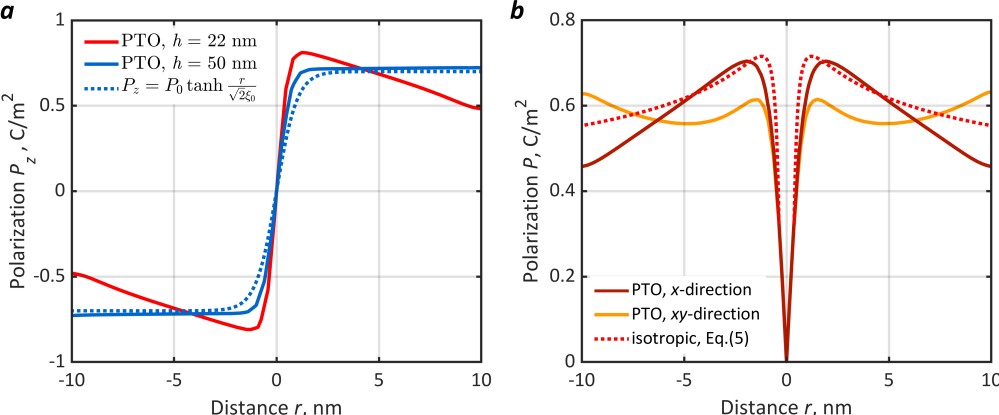

Figure 3: **Polarization distribution in $a$- and $c$-vortices.** (a) Distribution of $z$-component of polarization, $P_z$, in the cylinder with the DW formed by $a$-vortex, in the perpendicular to the DW direction. The solid blue and solid red lines correspond to the results of simulation for cylinders of $R = 10$ with $h = 50$ nm and $h = 22$ nm. The dotted blue line depicts the analytical dependence for the planar DW, $P_z = P_0 \tanh\left(r/\sqrt{2}\xi_0\right)$. (b) Radial distribution of the polarization magnitude in a $c$-vortex in the cylinder with $R = 10$ nm and $h = 6$ nm. The dark red and orange solid lines present the result of numerical simulations for the PTO model in $x$- and $xy$- directions with the minimal and maximal polarization overshoots, respectively. The red dotted line depicts the analytical approximation given by Eq. (5).

$h \simeq 6$ nm, which is the most symmetrical one. The shown in Fig. 3b radial distribution of polarization magnitude, $P(r)$, in different anisotropy directions demonstrates several remarkable features, coming from the long-range elastic interaction. After an initial increase from zero in the core-singular region, the dependence $P(r)$ passes through the maximum and then slowly decreases far from the core with no saturation at $r \to \infty$. This overshooting effect (also observed for $a$-vortices with $h \sim 2R$, Fig. 3a), is clearly seen in the 3D visualization plot with some anisotropy features (state V in Fig. 2). It was also obtained in atomistic simulations [16].

To gain an insight into such peculiar behaviour, we consider the model isotropic situation described by functional (2), assuming that a vortex has axial symmetry with the in-plane distributed polarization, presented in cylindrical coordinates $(\rho, \theta, z)$ as $\mathbf{P} = P(r)\hat{\boldsymbol{\theta}}$. The challenge in the calculation of the vortex properties is that the non-uniformly rotating polarization creates non-local elastic deformations that, in turn, affect the polarization itself. The self-consistent solution of this problem, given in Appendix A.2, presents the radial distribution of the polarization magnitude outside the vortex core as a series expansion over $\xi_0/R \ll 1$:

$$P(r) = P_0\left[\gamma_0\left(\frac{R}{r}\right)^{\frac{1-\mu}{2}} - \gamma_2\frac{\xi_0^2}{R^2}\left(\frac{R}{r}\right)^{\frac{\mu+3}{2}}\right], \qquad \mu^2 = 1 - \frac{\bar{q}_{44}}{4\bar{c}_{44}}\frac{4\bar{q}_{12}\bar{c}_{44} + \bar{q}_{44}\left(\bar{c}_{12} + 2\bar{c}_{44}\right)}{2\bar{a}_{11}^u\left(\bar{c}_{12} + \bar{c}_{44}\right) - \bar{q}_{12}^2}. \quad (5)$$

Remarkably, the dependence $P(r)$ has an exponential non-analytic form, unusual for the vortex-type systems in which the order parameter saturates far from the core. The first term in (5) corresponds to the long-range elastic contribution, and the second one describes the contribution of the gradient energy on approaching the vortex core region. The exponent-generating factor, $\mu$, is the function of the material parameters. In our isotropic model, $\mu \approx 0.67$, which ensures the decrease in the magnitude of both terms at large $r$. The dimensionless coefficients $\gamma_0$ and $\gamma_2$ are also expressed through the material parameters, as described in Appendix A.2. In our case they are estimated as $\gamma_0 \approx 0.79$, $\gamma_2 \approx 0.27$. The given

by Eq. (5) analytical dependence is presented in Fig. 3b by the red dotted curve. It correctly reproduces the main features of the numerical simulations: the non-monotonic dependence $P(r)$ and overshooting effect resulting from competition of elastic and gradient energy terms. The quantitative difference is associated with some ambiguity in selection of coefficients in functional (2), see Appendix A.2. Note, however, that although the range of parameters $0 < \mu < 1$ and $\gamma_0, \gamma_2 > 0$ we use is the most typical for oxide ferroelectrics, these constraints are not general, and other distinctive regimes may potentially occur.

With increasing disc size, the c-vortex patterns appear to be more complex than those observed in the cylinders with $R \simeq$ 6-10 nm and $h \simeq 6$ nm. In particular, for $R \simeq$ 10-14 nm at the same thickness, the vortex structure becomes deformed (state VI). The vortex-core region acquires the form of 180° DW, whereas the circular curvature of polarization lines concentrates inside the four 90° DWs, forming the rectangular flux-closure structure of polarization domains. For thicker discs, on approaching the transition to the a-vortex state at $h \sim R$, the flat 180° DW, stemming from the deformed vortex core, becomes screwed. It acquires the structure of a two-blade propeller with 90° twist of the blades, see state IV in Fig. 2. At larger radii, $R \gtrsim 14$ nm, and $h \simeq$ 6-12 nm, the system becomes unstable to the fragmentation on more vortices (state VII), predicted in [9]. The instability results from elastic strains, provided by the non-local contribution of the vortex core, see Eq. (5). The elastic tension produced by the central vortex becomes screened by the new oppositely winding vortices that enter from the disc side and create the multidomain state.

We present the energy of the c-vortex using the functional form similar to Eq. (4), namely:

$$E_c = \pi R^2 h F_c + \xi_0 R h F_{1c}, \tag{6}$$

where the first negative term, scaled as the disc cross-sectional area, corresponds to the bulk energy of the system with energy density $F_c < 0$. The absolute value of this parameter, $|F_c|$, is usually smaller than that for the uniform state $|F_0|$, and even than $|F_a|$, due to the significant elastic contribution. It results from the bulk stresses produced by matching the flux-closure 90° domains as well as by the long-range elastic energy of the deformed vortex core. The second term in (6), scaled as the disc radius, corresponds to the energy stored in the DWs. It is described by the effective energy density of DWs, $F_{1c} > 0$, which also accounts for the geometrical parameters of DWs. The fit of the phase-field simulation results for PTO discs (see Appendix A.3) gives the following numerical values of the parameters in Eq. (6): $F_c \approx -0.053 \times 10^9$ Jm$^{-3}$ and $F_{1c} \approx 0.78 \times 10^9$ Jm$^{-3}$. Notably, the vortex core deformations and even fragmentation of the system to the multivortex state do not significantly impact the extracted from the fitting parameters in Eq. (6) since the energies of all c-vortex states are very close to each other.

## 4.4 a- to c-vortex transition

Rotation of the vortex core axis from a- to c-direction upon the cylinder height decreasing occurs when $h$ becomes comparable to $R$. To estimate the critical geometrical parameters, we compare the a- and c-vortex energies, plotted in Fig. 4 as a function of $h$ by the black dots. The families of the red and blue lines correspond to the fits of the numerical data for the c- and a- vortices given by relations (4) and (6), respectively. Further details on the calculation of the energies are given in Appendix A.3.

The corresponding transition line, depicted in the R-h phase diagram (Fig. 2) by the solid blue curve, is given by the relation $E_a(h,R) = E_c(h,R)$. This line approximates the transition from a- to c-vortex states, illustrated by the warm and cool colours in the phase diagram in Fig. 2, as observed in the simulations. The slight shift of the blue line with respect to experimental transition is due to the more complex structure of the vortex cores in the transition region, compared to the analytical model.

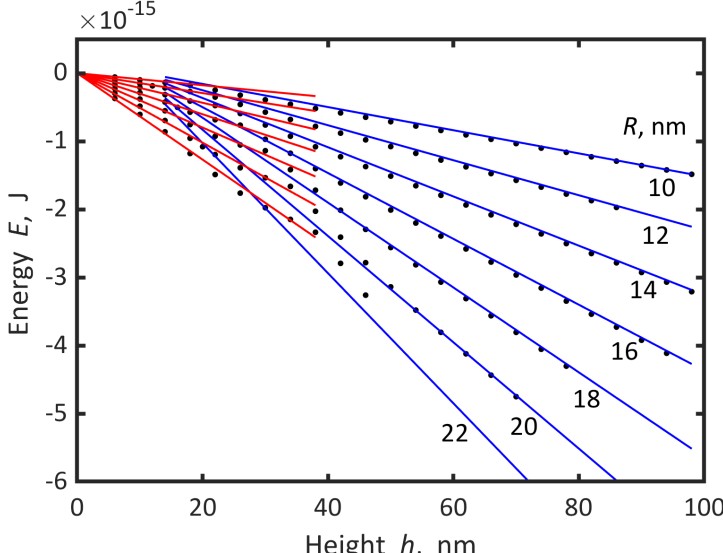

Figure 4: **Energies of the vortex states**. Black points correspond to the numerical data obtained from the phase-field simulations. Red and blue lines correspond to the best-fit results for the $c$- and $a$-vortices in cylinders with different $R$.

## 4.5  Evolution with temperature

Our simulations show that the $c$-vortex phase emerges on cooling from the paraelectric state in cylinders at a critical temperature $T_{cc}$. This temperature is lower than the bulk PTO transition temperature $T_c = 752\,\text{K}$ and depends on the cylinder radius, but not on its height. However, for cylinders with $h \gtrsim R$ the $a$-vortex phase becomes stable (*i.e.* possesses the smaller energy than the $c$-vortex phase) at temperature $T_{ca}^* < T_{cc}$. The exemplary temperature-height phase diagram illustrating the corresponding sequence of transitions is shown in Fig. 5a for the cylinder of radius $R = 12\,\text{nm}$. The temperature dependence of the average polarization amplitude, $P(T)$, is shown in Fig. 5b for the cylinder of the same radius and different heights $h = 6\,\text{nm}$, $h = 14\,\text{nm}$, $h = 18\,\text{nm}$, and $h = 26\,\text{nm}$. The discontinuities in the $P(T)$ dependencies correspond to the $c$- to $a$- vortex transition on heating. It takes place at superheating transition temperature, $T_{ca}^+$, shown in Fig. 5a by the dotted blue line. Note that on cooling, the $c$-vortex phase does not transit to the $a$-vortex phase, staying in the metastable state, albeit their energy balance becomes positive below $T_{ca}^*$.

To describe the temperature-driven $c$- to $a$-vortex state transition, we linearize GL equations, obtained from the isotropic functional (2) in the vicinity of the transition to the paraelectric state, neglecting the elastic terms (that are of order $\sim P^3$) and taking into account the divergenceless structure of the polarization field:

$$a_1(T)\mathbf{P} = \bar{G}\nabla^2\mathbf{P}, \qquad \text{div}\,\mathbf{P} = 0. \tag{7}$$

Eq. (7) is solved with free boundary conditions $(\mathbf{n}\nabla)\mathbf{P} = 0$ and with the constraint of zero depolarization charge at the surface, $(\mathbf{n}\mathbf{P}) = 0$, where $\mathbf{n}$ is a unit vector normal to the boundary.

Next, we compare the obtained instability temperatures of transition from the paraelectric phase to $c$- and $a$-vortex states, see Appendix A.4. The linearized solutions for the polarization distribution, written in the cylindrical coordinates $(r, \theta, z)$ for the competing $a$-vortex and $c$-vortex phases, and the reduction of corresponding critical temperatures $T_{cc}$ and $T_{ca}$ with

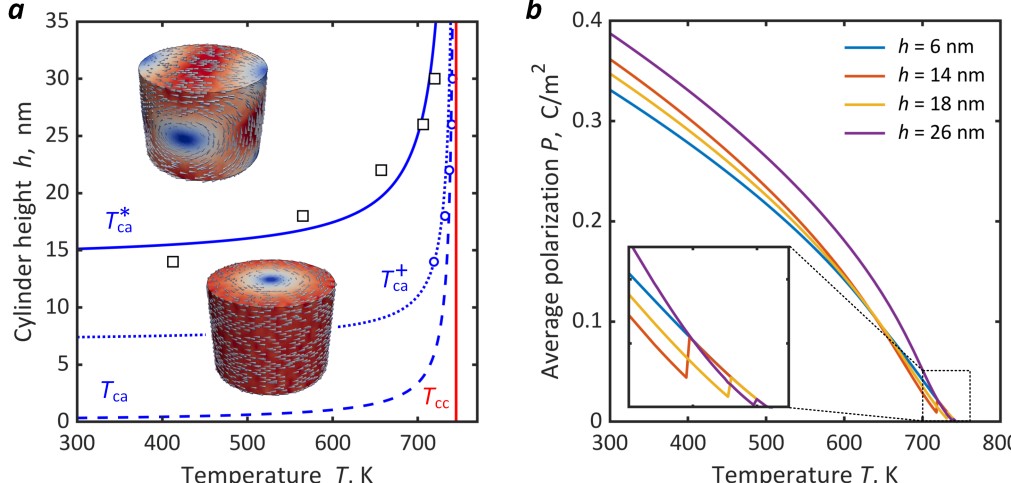

Figure 5: **Temperature-dependent properties of vortex states.** (a) The $T$-$h$ diagram of vortex states emerging in cylinders with $R = 12$ nm. The transition from the paraelectric to the $c$-vortex state (lower cylinder) occurs at temperature $T_{cc}$, shown by the solid red line. This state remains stable down to the room temperature for cylinders with $h \lesssim 15$ nm. For longer cylinders the $a$-vortex state becomes more stable below thermodynamic critical temperature $T^*_{ca}$ shown by the solid blue line. The dotted blue line, $T^+_{ca}$, shows the critical temperature at which the $a$- to $c$-vortex transition occurs on heating. The auxiliary dashed blue line, $T_{ca}$, demonstrates the paraelectric-to-$a$-vortex virtual instability of the linearized GL equation, that is the precursor of the observed $a$- to $c$-vortex transition. (b) Temperature dependence of the average polarization magnitude for cylinders with different heights and $R = 12$ nm. The jumps of polarization close to the transition temperature (magnified at the bottom left corner) correspond to the $a$- to $c$-vortex state transition on heating at temperatures $T^+_{ca}$.

respect to $T_c$ are given by:

$$c\text{-vortex:} \quad \mathbf{P}_c = C_c(T_{cc} - T)^{1/2} J_1\left(\lambda_1 \frac{r}{R}\right) \hat{\boldsymbol{\theta}}, \qquad T_{cc} = T_c\left(1 - \lambda_1^2 \frac{\xi_0^2}{R^2}\right),$$

$$a\text{-vortex:} \quad \mathbf{P}_a = C_a(T_{ca} - T)^{1/2} J_1\left(\lambda_1 \frac{r}{R}\right)\cos\theta \, \hat{\mathbf{z}}, \qquad T_{ca} = T_c\left(1 - \lambda_1^2 \frac{\xi_0^2}{R^2} - 2\kappa_a \frac{\xi_0}{h}\right). \quad (8)$$

Here, $J_1$ is the first-order Bessel function, $\lambda_1 = 1.8412$ is the first zero of the derivative $\partial_x J_1(x)$, and coefficients $C_c$ and $C_a$ are found by solving the nonlinear equations. The last term in $T_{ca}$ is included to account for the additional electrostatic energy and disturbance of the order parameter at the cylinder edges due to the emergence of the depolarization field. Such effect is absent in the case of $c$-vortex. Importantly, this electrostatics-driven contribution (scaled as $\xi_0$, with material-dependent dimensionless coupling parameter $\kappa_a$) reduces $T_{ca}$ with respect to $T_{cc}$, always stabilizing the $c$-vortex state just below the transition from the paraelectric phase. The emergence of the $c$-vortex before the $a$-vortex state either renormalizes the $a$-vortex transition temperature to $T^*_{ca} < T_{ca}$ or completely pushes it out, as shown in Fig. 5a. In the first case, occurring at $h \gtrsim R$, we observe the $a$-vortex state at room temperature, whereas in the second scenario, for $h \lesssim R$, the $c$-vortex state persists until the room temperature.

# 5 Conclusion

In the current work, we performed the complete study of the topological polarization states confined in free-standing cylindrical nanoparticles or nanorods of PTO for the extended range of geometry parameters and temperatures. Having revealed a large variety of swirling polarization textures, we demonstrate that, on the topological level, they can be all classified as vortex states with differently oriented vortex axes. Some of these textures are similar to domains and vortices, obtained by phase-field simulations for long cylinders and nanowires of BaTiO$_3$ embedded in the external media [13] and with the account of the flexoelectric effect [14]. They are also similar to the topological structures observed by atomic-level simulations in nanoparticles of PTO [16] at the smaller scale of 4-10 nm, although some quantitative differences take place, presumably due to the interface effects. The diversity of vortex manifestations highlights the universality of the topological structures in confined ferroelectrics. However, a more detailed investigation of the multi-scale matching of the Ginzburg-Landau phase-field modeling and atomic-level simulations and comparison with experiment is required to identify the scale at which the discrete atomic structure and the influence of the surface effects, for instance, the surface tension, become significant.

On a more general perspective, topological classification of the divergenceless fields in confined geometries, introduced by Arnold in relation to studies on incompressible liquid streams [29], suggests that globally the system splits into the cells of elementary topological excitations of either of two types – conventional 2D vortices and more elaborated 3D knotted structures, Hopfions, in which the polarization lines escape in the third dimension from the singular core. Both vortices [1–3] and Hopfions [30] have been discovered in nanostructured ferroelectrics, in particular, in confined geometries of thin films, superlattices, nanodots, and nanoparticles.

Note, however, that although the appearance of the vortex states in the cylinder complies with the Arnold theorem, we did not observe the emergence of the Hopfion state in our simulations. This happens due to the strong anisotropy of PTO and is in line with other simulations in this material [16, 31]. While the discussed geometric configuration allows Hopfion states, anisotropic contribution makes them energetically unfavourable. In this respect, it would be interesting to investigate the polarization textures in nanocylinders of the less anisotropic material, Pb$_x$Zr$_{1-x}$TiO$_3$, usually hosting Hopfions in spherical nanoparticles [30]. This study is currently in progress.

## Acknowledgements

The authors would like to thank Franco Di Rino and Marcelo Sepliarsky for discussions and helpful comments.

**Funding information.** S.K. acknowledges the support from the Alexander von Humboldt Foundation and from the EU H2020-MSCA-RISE-MILEAGE action, project number 734931. The research of I.L., M.P, and A.R. was funded by the EU H2020-MSCA-RISE-MELON action, project number 872631. The research of Y.T. was supported by the EU H2020-MSCA-ITN-MANIC action, project number 861153.

**Author contributions.** Conceptualization, S.K., I.L.; numerical simulations, M.P, Y.T., A.R.; theory, I.L, S.K.; data analysis, S.K., I.L., M.P, A.R., Y.T.; manuscript writing, S.K., I.L.; project administration, S.K., A.R. All authors have read and agreed to the published version of the manuscript.

**Data availability.** The generated data are available from the corresponding author upon reasonable request.

## A Supplementary Information

### A.1 Computational techniques

Numerical calculations were performed by using the phase-field method, implemented on the FEniCS computing platform [32]. The nonlinear part of relaxation differential equations is coming from the variation of the free energy functional (1) with respect to the polarization **P**:

$$-\gamma \frac{\partial \mathbf{P}}{\partial t} = \frac{\delta \mathcal{F}^{\mathrm{u}}}{\delta \mathbf{P}}. \tag{A.1}$$

Electrostatic potential $\varphi$ and elastic strains $u_{ij}$ are obtained as the result of the solution of the linear equations:

$$\varepsilon_0 \varepsilon_b \nabla^2 \varphi = \partial_i P_i, \tag{A.2}$$

$$c_{ijkl} \partial_j u_{kl} - q_{ijkl} \partial_j P_k P_l = 0. \tag{A.3}$$

The parameter $\gamma$ determines the time scale for computations. Its value is irrelevant for the calculation of the static structures and is taken equal to 1 ns.

Discretization of computational space into tetrahedrons was performed with 3D mesh generator gmsh [33], see Fig. 6. The yellow cylinder represents the volume of ferroelectric, $\Omega_c$. The full surface of $\Omega_c$, denoted as $\partial \Omega_c$, includes top, bottom, and side surfaces. The ferroelectric cylinder is embedded in the vacuum cylindrical volume $\Omega_m$ with surface $\partial \Omega_m$. In Fig. 6 a half of this volume is shown in transparent emerald. To accelerate the computations, the density of the finite element grid was set to decrease from $\partial \Omega_c$ to $\partial \Omega_m$. We apply the Dirichlet boundary condition $\varphi = 0$ at $\partial \Omega_m$, while other boundary conditions are free.

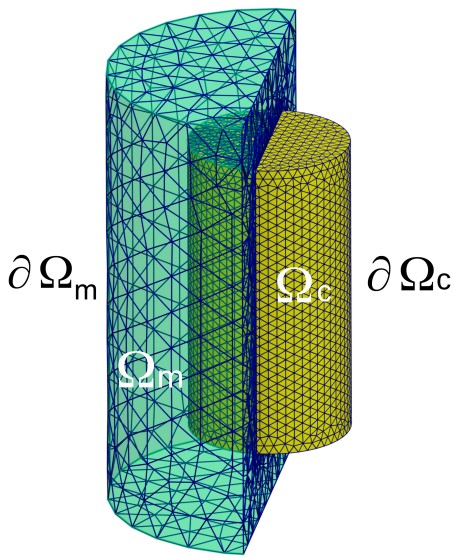

Figure 6: **Finite element mesh of a ferroelectric cylinder with the surrounding medium.** The yellow cylinder corresponds to ferroelectric volume $\Omega_c$ with surface $\partial \Omega_c$. The transparent emerald half-cylinder represents a part of the surrounding medium volume $\Omega_m$ with the surface $\partial \Omega_m$.

For numerical calculations we used the standard coefficients for PTO [26, 34] $a_1 = 3.8 \times 10^5 (T - 752\,\text{K}) C^{-2} m^2 N$, $a_{11}^\sigma = -0.73 \times 10^8 C^{-4} m^6 N$, $a_{12}^\sigma = 7.5 \times 10^8 C^{-4} m^6 N$, $a_{111} = 0.26 \times 10^9 C^{-6} m^{10} N$, $a_{112} = 0.61 \times 10^9 C^{-6} m^{10} N$, $a_{123} = -3.7 \times 10^9 C^{-6} m^{10} N$, $G_{1111} = 2.77 \times 10^{-10} C^{-2} m^4 N$, $G_{1122} = 0$, $G_{1212} = 1.38 \times 10^{-10} C^{-2} m^4 N$, $q_{1111} = 15.54 \times 10^9 C^{-2} m^2 N$, $q_{1122} = -2.06 \times 10^9 C^{-2} m^2 N$, $q_{1212} = 3.75 \times 10^9 C^{-2} m^2 N$, $c_{1111} = 174.6 \times 10^9 m^{-2} N$, $c_{1122} = 79.36 \times 10^9 m^{-2} N$, $c_{1212} = 111.1 \times 10^9 m^{-2} N$. The coefficients $a_{11}^u$ and $a_{12}^u$ are calculated from $a_{12}^\sigma$ and $a_{12}^\sigma$ using the procedure described in [27].

The variable time BDF2 stepper [35] was used to approximate the time derivative on the left hand side of functional variation (A.1). We chose random distribution of **P** vector components with average amplitude $\sim 10^{-6}\,\text{C m}^{-2}$ in $\Omega_c$ at the first time step of simulation to model the relaxation from initial paraelectric phase. Solution of nonlinear equations coming from free energy functional (A.1) variation with respect to **P** is performed with the Newton based nonlinear solver with line search and generalized minimal residual method with restart [36, 37]. Linear systems that come from discretization of Poisson (A.2) and elasticity (A.3) equations were solved separately using a generalized minimal residual method with restart.

## A.2  Structure of an isotropic vortex

To reproduce analytically the features of the polarization distribution in a strain-coupled axisymmetric vortex state, we employ a simplified isotropic form of the free energy functional (2), in which the effective electrostrictive, $\bar{q}_{ij}$, and elastic, $\bar{c}_{ij}$, coefficients are obtained by isotropization of the corresponding tensors, that for the general 4-th rank tensor $T_{ijkl}$ in cubic crystal reads as [38, 39] $\bar{T}_{ijkl} = \alpha \delta_{ij} \delta_{kl} + \beta \left( \delta_{ik} \delta_{jl} + \delta_{il} \delta_{jk} \right)$ with $\alpha = \frac{1}{5} \left( T_{1111} + 4 T_{1122} - 2 T_{1212} \right)$ and $\beta = \frac{1}{5} \left( T_{1111} - T_{1122} + 3 T_{1212} \right)$. Then, the corresponding coefficients in functional (2) are $\bar{c}_{12} = \bar{c}_{1122} = 54.0 \times 10^9\,\text{m}^{-2}\text{N}$, $\bar{c}_{44} = \bar{c}_{1212} = 85.7 \times 10^9\,\text{m}^{-2}\text{N}$, $\bar{q}_{12} = \bar{q}_{1122} = 1.15 \times 10^9 C^{-2} m^2 N$, and $\bar{q}_{44} = 2\bar{q}_{1212} = 8.87 \times 10^9\,C^{-2} m^2 N$. In addition, the sixth-order polarization terms were neglected in the GL part of the functional and the averaged coefficients were expressed through the PTO coefficients in (1) in a way to preserve the principal characteristics of the material. We take $\bar{a}_{11}^u = 0.41 \times 10^9\,C^{-4} m^6$ and $a_1(T)$ as in (1) to have the same critical temperature, $T_c = 752\,\text{K}$, and polarization magnitude $P_0 = \left( |a_1|/2\bar{a}_{11}^\sigma \right)^{1/2} \simeq 0.7\,\text{C m}^{-2}$ at room temperature; here, $\bar{a}_{11}^\sigma = \bar{a}_{11}^u - \frac{\bar{c}_{12} \bar{q}_{44}^2 + \bar{c}_{44} \left( \bar{q}_{12} + \bar{q}_{44} \right)^2 + 2 \bar{c}_{44} \bar{q}_{12}^2}{2 \bar{c}_{44} \left( 3 \bar{c}_{12} + 2 \bar{c}_{44} \right)}$. The gradient energy in PTO with coefficients from [34] has already an isotropic form, hence we take $\bar{G} = G_{1212} = 1.38 \times 10^{-10}\,C^{-2} m^4 N$.

We derive now the structure of the axisymmetric vortex in frame of the isotropic model. In the cylindrical coordinates $(r, \theta, z)$, the functional (2) with the axisymmetric polarization distribution, $\mathbf{P}(r) = (0, P, 0)$, corresponding to the $c$-vortex, is written as:

$$\mathcal{F}_{iso}^u = a_1 P^2 + \bar{a}_{11}^u P^4 + \bar{G} \left[ (\partial_r P)^2 + (P/r)^2 \right] - \bar{q}_{12} P^2 \left( u_{rr} + u_{\theta\theta} + u_{zz} \right) - \bar{q}_{44} P^2 u_{\theta\theta}$$
$$+ \frac{1}{2} \bar{c}_{12} \left( u_{rr} + u_{\theta\theta} + u_{zz} \right)^2 + \bar{c}_{44} \left( u_{rr}^2 + u_{\theta\theta}^2 + u_{zz}^2 \right),$$

where the strain tensor components are $u_{rr} = \partial_r u_r$, $u_{\theta\theta} = u_r/r$, $u_{zz} = \partial_z u_z$.

The variation of the energy functional with respect to variables $P$, $u_r$, $u_z$ provides three equations:

$$\bar{G} \left( \nabla_r^2 P - P/r^2 \right) = a_1 P + 2 \bar{a}_{11}^u P^3 - \bar{q}_{12} P \left( \partial_r u_r + u_r/r + u_{zz} \right) - \bar{q}_{44} P u_r/r, \qquad (\text{A.4})$$
$$\left( \bar{c}_{12} + 2 \bar{c}_{44} \right) \left( \nabla_r^2 u_r - u_r/r^2 \right) = -\bar{q}_{44} P^2/r + \bar{q}_{12} \partial_r \left( P^2 \right) - \bar{c}_{12} \partial_r u_{zz},$$
$$\left( \bar{c}_{12} + 2 \bar{c}_{44} \right) u_{zz} = \bar{q}_{12} P^2 - \bar{c}_{12} \left( \partial_r u_r + u_r/r \right);$$

and two boundary conditions:

$$(\partial_r P)_{r=R} = 0, \qquad \left[ \bar{c}_{12} \left( \partial_r u_r + u_r/r + u_{zz} \right) + 2 \bar{c}_{44} u_{rr} - \bar{q}_{12} P^2 \right]_{r=R} = 0.$$

Here, $\nabla_r^2 = \partial_r^2 + r^{-1}\partial_r$.

To solve approximately equations (A.4), we consider first the long-range asymptotic behaviour of polarization (at scales larger than $\xi_0$), assuming $\bar{G} = 0$. The corresponding solution for the polarization, $\widetilde{P}(r)$, is obtained as:

$$\widetilde{P}^2 = \widetilde{P}_0^2 \left(\frac{r}{R}\right)^{\mu-1}, \qquad \mu^2 = 1 - \frac{\bar{q}_{44}}{4\bar{c}_{44}} \frac{4\bar{q}_{12}\bar{c}_{44} + \bar{q}_{44}\left(\bar{c}_{12} + 2\bar{c}_{44}\right)}{2\bar{a}_{11}^{u}\left(\bar{c}_{12} + \bar{c}_{44}\right) - \bar{q}_{12}^2}, \tag{A.5}$$

where the constant $\widetilde{P}_0$ is found from the boundary conditions:

$$\widetilde{P}_0^2 = -\frac{a_1}{2\bar{a}_{11}^{u} - \bar{q}_{12}\frac{\bar{q}_{12}(2\mu+1)+\bar{q}_{44}}{\bar{c}_{12}(2\mu+1)+2\mu\bar{c}_{44}}} \frac{2(\mu+1)\bar{q}_{12}\bar{c}_{44} + \bar{q}_{44}\left(\bar{c}_{12} + 2\bar{c}_{44}\right)}{4\bar{q}_{12}\bar{c}_{44} + \bar{q}_{44}\left(\bar{c}_{12} + 2\bar{c}_{44}\right)} \frac{3\bar{c}_{12} + 2\bar{c}_{44}}{(2\mu+1)\bar{c}_{12} + 2\mu\bar{c}_{44}}.$$

The numerical values of the constants are estimated as $\mu = 0.67$ and $\widetilde{P}_0 = 0.56$ Cm$^{-2}$. Note that the natural boundary condition $(\partial_r P)_{r=R} = 0$ is fulfilled in the long-range approximation since the polarization derivative at $r = R$ scales as $\sim \xi_0/R$ thus approaching zero.

Now, we calculate $P(r)$ from equations (A.4) more exactly in the next order in $\xi_0/R$. Presenting $P(r)$ as

$$P(r) = \widetilde{P}(r) + p(r), \tag{A.6}$$

we obtain

$$p = \widetilde{P}_0 \frac{\sqrt{\pi}}{2}\left(\frac{1}{2} - \nu\right)\Gamma\left(\frac{1}{2} - \nu\right)\left(\frac{\beta}{2}\right)^{\nu-1}\left[I_\nu\left(\beta\left(\frac{r}{R}\right)^{1/\nu}\right) - L_{-\nu}\left(\beta\left(\frac{r}{R}\right)^{1/\nu}\right)\right],$$

where $I_\nu(x)$ and $L_\nu(x)$ are the modified Bessel and modified Struve functions of order $\nu$, respectively, $\Gamma(x)$ is the gamma function, and the following notations are used: $\nu = 2/(\mu+1)$, $\beta = \nu\widetilde{P}_0\sqrt{-a_{11}^*/a_1}\,R/\xi_0$, and $a_{11}^* = \bar{a}_{11}^{u} - \bar{q}_{12}^2/\left(2\bar{c}_{12} + 4\bar{c}_{44}\right)$.

The asymptotic expansion of the special functions $I_\nu(x)$ and $L_\nu(x)$ [40,41] gives the long-range approximation (5) of the total solution (A.6) with coefficients

$$\gamma_0 = \frac{\widetilde{P}_0}{P_0}, \qquad \gamma_2 = -\frac{a_1}{2a_{11}^*}\frac{(3-\mu)(\mu+1)}{8\widetilde{P}_0 P_0}.$$

Close to the vortex core, the polarization linearly depends on the distance as

$$P \propto \widetilde{P}_0 \frac{\sqrt{\pi}}{2}\frac{\left(\frac{1}{2} - \nu\right)\Gamma\left(\frac{1}{2} - \nu\right)}{\nu\Gamma(\nu)}\left(\frac{\beta}{2}\right)^{2\nu-1}\frac{r}{R}.$$

The comparison of the given by the Eq. (A.6) analytical dependence $P(r)$ (dotted red line) with results of the numerical simulations (solid red line) is shown in Fig. 7a for the cylinder with $R = 10$ nm and $h = 6$ nm. The result captures the functional behaviour of polarization in a strain-coupled $c$-vortex state and matches well the simulation results in the whole interval of $r$, including the core.

The long-range asymptotic solution for the elastic tensor components obtained from (A.4) has the following form:

$$\begin{aligned}
\widetilde{u}_{rr} &= C_1\mu\left[2\bar{a}_{11}^{u}\left(\bar{c}_{12} + 2\bar{c}_{44}\right) - \bar{q}_{12}^2\right]\widetilde{P}^2 + C_2 a_1\left(\bar{c}_{12} + 2\bar{c}_{44}\right), \\
\widetilde{u}_{\theta\theta} &= C_1\left[2\bar{a}_{11}^{u}\left(\bar{c}_{12} + 2\bar{c}_{44}\right) - \bar{q}_{12}^2\right]\widetilde{P}^2 + C_2 a_1\left(\bar{c}_{12} + 2\bar{c}_{44}\right), \\
\widetilde{u}_{zz} &= C_1\left[(\mu+1)\left(\bar{q}_{12}^2 - 2\bar{c}_{12}\bar{a}_{11}^{u}\right) + \bar{q}_{44}\bar{q}_{12}\right]\widetilde{P}^2 - 2C_2 a_1\bar{c}_{12},
\end{aligned} \tag{A.7}$$

in which $\widetilde{P}^2$ is given by Eq.(A.5), $C_1^{-1} = 2(\mu+1)\bar{q}_{12}\bar{c}_{44} + \bar{q}_{44}\left(\bar{c}_{12} + 2\bar{c}_{44}\right)$ and $C_2^{-1} = 4\bar{c}_{44}\bar{q}_{12} + \bar{q}_{44}\left(\bar{c}_{12} + 2\bar{c}_{44}\right)$. Fig. 7b demonstrates good agreement between the numerically and analytically calculated dependencies $u_{rr}(r)$ and $u_{zz}(r)$ outside the vortex core.

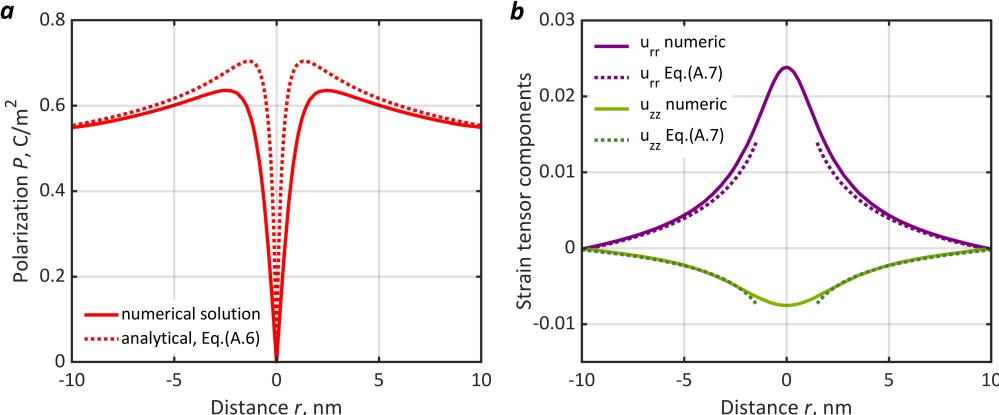

Figure 7: **Isotropic model for $c$-vortex.** (a) Radial distribution of polarization in $c$-vortex in the cylinder with $R = 10$ nm and $h = 6$ nm. The solid red line presents the result of numerical simulation, the red dotted line corresponds to the results of analytical calculations given by Eq. (A.6). (b) Elastic tensor components as a function of distance from the vortex core. The solid purple and solid green lines depict the numerical solution for components $u_{rr}$ and $u_{zz}$, respectively; the dotted purple and dotted green lines demonstrate the analytical approximation given by (A.7).

## A.3 Energy parameters

Here, we find the numerical parameters in the approximate expressions for the energies of vortices, given in Section 4.2-4.3. The obtained from the phase-field simulations energies of vortex states in cylinders with different $R$ are shown in Fig. 4 (Section 4.4) as a function of $h$ by the black dots. Red and blue lines fit the numerical data for the $c$- and $a$-vortices according to equations (6) and (4), respectively. The following best-fit parameters were obtained: $F_c \approx -0.053 \times 10^9 \text{Jm}^{-3}$, $F_{1c} \approx 0.78 \times 10^9 \text{ Jm}^{-3}$ for the $c$-vortex states and $F_a \approx -0.07 \times 10^9 \text{Jm}^{-3}$, $F_{1a} \approx 0.50 \times 10^9 \text{ Jm}^{-3}$, and $F_æ \approx 0.29 \times 10^9 \text{ Jm}^{-3}$ for the $a$- vortex states.

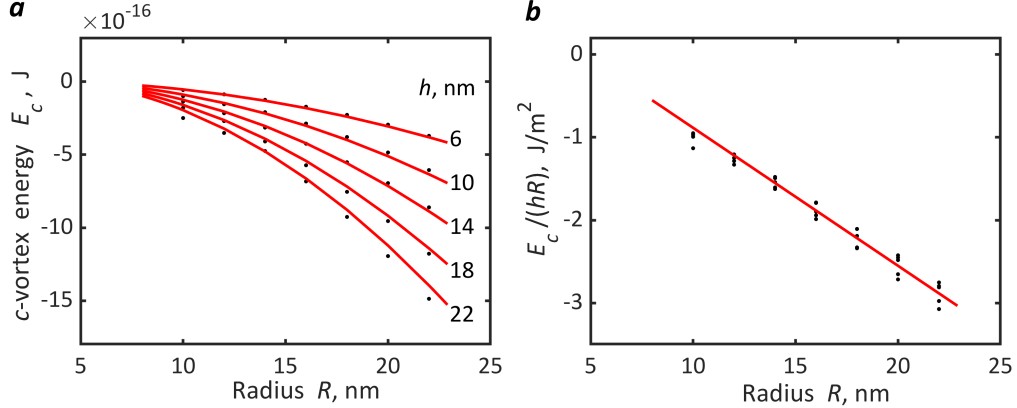

Figure 8: **Energy fit for $c$-vortex.** (a) Numerical data and the best fit for the $c$-vortices in cylinders with different $h$. (b) Linear scaling of the parameter $E_c/(hR)$.

Figures 8 and 9 demonstrate the details of the fitting of the energies $E_c$ and $E_a$ of the $c$- and $a$- vortex states, respectively. The numerical data for vortex energies in cylinders with different $h$, and their fits according to equations (6) and (4) are shown as function of $R$ in

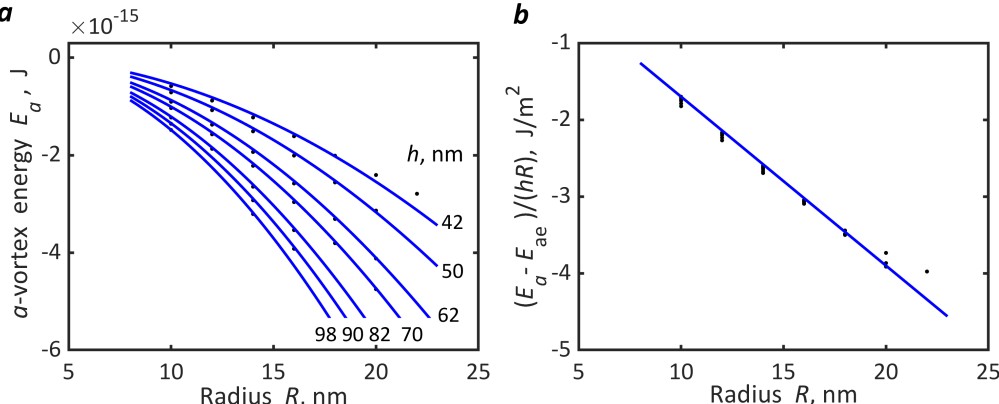

Figure 9: **Energy fit for $a$-vortex.** (a) Numerical data and the best fit for the $a$-vortices in cylinders with different $h$. (b) Linear scaling of the parameter $(E_a - E_æ)/(hR)$, in which $E_æ = 2\pi R^2 \xi_0 F_æ$ is the vortex terminal energy.

panels 8a and 9a, respectively. Panels 8b and 9b demonstrate the data combinations, $E_c/(hR)$ and $(E_a - E_æ)/(hR)$ (with the terminal energy $E_æ = 2\pi R^2 \xi_0 F_æ$), which according to the equations (6) and (4) scale as linear functions, allowing for the accurate determination of the fitting parameters. The scattered points in Fig. 9 correspond to the a-vortex states in cylinders with $h = 42$ nm and radii $R \gtrsim 20$ nm, which is close to region II (twisted DW state) in diagram in Fig 2. Due to slight distortion of the vortex core in this transient region the linear assumption given by (4) can deviate.

## A.4 Critical temperatures

Linearized GL equations for the divergenceless polarization field, $\mathbf{P} = P_\theta(r)\,\hat{\boldsymbol{\theta}} + P_z(r,\theta)\,\hat{\mathbf{z}}$, are written in cylindrical coordinates as

$$\bar{G}\left(\nabla^2 P_\theta - \frac{P_\theta}{r^2}\right) = \alpha_1(T - T_c)P_\theta,\tag{A.8a}$$

$$\bar{G}\nabla^2 P_z = \alpha_1(T - T_c)P_z,\tag{A.8b}$$

where $\nabla^2 = r^{-1}\partial_r(r\partial_r) + r^{-2}\partial_\theta^2$. We assume here that the polarization distribution is uniform along $z$-direction and look for the solution of Eqs. (A.8) with the natural boundary conditions $\partial_r P_\theta(R) = 0$, $\partial_r P_z(R) = 0$.

Besides the uniform monodomain $c$-phase with $\mathbf{P}_u = C_u(T_c - T)^{1/2}\,\hat{\mathbf{z}}$ and critical temperature $T_c$, there are two competitive solutions of Eqs. (A.8).

(i) The solution of Eq. (A.8a) $\mathbf{P}_c = C_c(T_{cc} - T)^{1/2}J_1(\lambda_1 r/R)\,\hat{\boldsymbol{\theta}}$ corresponds to the $c$-vortex,

(ii) and the solution of Eq. (A.8b) $\mathbf{P}_a = C_a(T_{ca} - T)^{1/2}J_1(\lambda_1 r/R)\cos\theta\,\hat{\mathbf{z}}$ is the precursor of the vertical domain wall which forms the $a$-vortex state.

Here, $J_1$ is the first-order Bessel function, $\lambda_1 = 1.8412$ is the first zero of the derivative $\partial_x J_1(x)$. The corresponding critical temperatures, $T_{cc} = T_c(1 - \lambda_1^2 \xi_0^2/R^2)$ and $T_{ca} = T_c(1 - \lambda_1^2 \xi_0^2/R^2)$, are found as eigenvalues of Eqs. (A.8). They appear to be equal; however, the accounting of the depolarization terminal energy of the $a$-vortex makes $T_{ca}$ smaller than $T_{cc}$ as discussed in Section 4.5. The coefficients $C_u$, $C_a$ and $C_c$ are found from the solution of the nonlinear problem.

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
