# Peer review of "Vortex states in a PbTiO$_3$ ferroelectric cylinder"

_SciPost Physics, doi:SciPost Phys. 14, 056 (2023)_

## Round 1 · Referee Report · Anonymous (Referee 5) · 2022-7-7

Strengths
1- interest and timeliness in view of current attention to topological defects in ferroelectrics
2-Usefulness for analysis of experimental data
Weaknesses
1-no particular weaknesses identified, some changes are requested, see below
Report
In view or recent developments in the field the proposed manuscript is topical and timely. It offers a detailed analytical and numerical description of vortex structures that may appear in ferroelectric cylindrical nanodots and nanowires due to depolarization effects. The authors take into account elastic and electrostatic energy contributions and demonstrate their role in the formation of polar structures, revealing non-trivial features in polarization distribution.
Employing phenomenological approach together with numerical phase-field simulations, the authors classify a variety of vortex-like textures in nanoscale cylinders and build the corresponding phase diagram as a function of the radius and height of cylinders, as well as of the temperature. The resulting systematic description is of practical use in the experimental design of ferroelectric-based nanoelectronic elements. Overall, the manuscript is well-written, and clearly presents the results and relevant discussions. I do recommend the manuscript for publication in SciPost Physics after addressing several technical issues. Specifically:
1) Equation (5) gives the polarization distribution of an isotropic vortex in the cylindrical nanodot. However, this expression would not fulfill the natural boundary condition of zero polarization derivatives at r=R. Comment on this would be useful in the manuscript.
2) In numerical simulations, the ferroelectric cylinders are surrounded by non-ferroelectric medium as shown in Fig.5. In the text, the cylinders are described as free-standing particles. I think it would be practical to mention the requirements for this surrounding media needed to consider the cylinders as free-standing, as well as to discuss the possible technological realizations of such free-standing nanostructures.
Requested changes
1- Equation (5) gives the polarization distribution of an isotropic vortex in the cylindrical nanodot. However, this expression would not fulfill the natural boundary condition of zero polarization derivatives at r=R. Comment on this would be useful in the manuscript.
2- In numerical simulations, the ferroelectric cylinders are surrounded by non-ferroelectric medium as shown in Fig.5. In the text, the cylinders are described as free-standing particles. I think it would be practical to mention the requirements for this surrounding media needed to consider the cylinders as free-standing, as well as to discuss the possible technological realizations of such free-standing nanostructures.
Anonymous on 2022-05-18 [id 2492]
In this paper, the authors report that the orientation of the vortex core in PbTiO3 cylinder could be tuned by the geometrical parameters and temperature using phase-field simulations. This work is systematic and would provide the guidance for designing the polar topological states experimentally in ferroelectric nanostructures. Following are the comments and questions.
1. The authors stated that the basic idea of the polarization vortices formation is sketched in Fig. 1. Under short-circuited boundary conditions, the uniform polarization occurs in the PbTiO3 cylinder, while the vortex states are observed in the PbTiO3 cylinder under open-circuited boundary conditions. However, as reported by Li S. et al. (Appl. Phys. Lett. 111, 052901 (2017)), the symmetry of the electrical boundary conditions played more crucial influences on the formation of flux-closure domains. Thus, additional theoretical results or discussions are suggested to be included in this paper to illuminate the formation and transition of vortex states in ferroelectric cylinders.
2. Figure 7 is suggested to be included in the main text, which would facilitate to understand the transition between a-vortex states and c-vortex states displayed in the phase diagram in Fig. 2.
3. About the title, “in a ferroelectric cylinder” is suggested to be more specific, as in the manuscript, only PbTiO3 cylinder was discussed.

---

## Round 2 · Referee Report · Anonymous (Referee 2) · 2022-9-19

Report
This paper present a complete theoretical study of the topological polarization states in free-standing PbTiO3 cylinders, as a function of their length, radius, and temperature, resulting in a detailed phase diagram showing the transition from c-phase to the formation of a- or c-vortices or the composition of the two, arising from the competition of the elastic and electrostatic interactions. The calculations are based on the free energy density functional, simplified for the analytical calculations or using the phase-field numerical modeling, resulting in comparable results.
The topic is particularly timely, as the interest of the community in exotic polarisation textures and topologies in lead titanate has never been higher, with recent reports of skyrmions, vortices, and bubble domains. The results presented here will lead to a better theoretical understanding of this technologically important material, allowing a more systematic control over its more complex polarisation topologies, and paving the way for many future discoveries.
The paper is well written and accessible to theoreticians as well as experimentalists.
For these reasons, I recommend the paper to be published with only minor changes/suggestions.
-
In the case of a-vortices, how is the orientation of the in-plane vortex-axis determined? Is itartificially aligned with the [100] or [010] axis or can it take any other in-plane direction?
-
p.4 When stating that the c-state occurs only in very long cylinders with h>500nm: shouldn't it also vary as a function of R?
-
p.5 Figure 2 The solid blue curve visible on the phase diagram is explained only later (p.9, Fig.4 + text) - this information might be added already in the caption of Figure 2.
-
p.6 The authors mention that "the observed perturbed polarization texture at cylinder edges possesses the spontaneously broken chirality, the effect that was thought to occur due to the flexoelectric contribution" - it would have been nice if the authors extended a bit this very interesting discussion. Indeed from Fig. 7, one sees that the strain gradients are very large, therefor one expects an important role played be flexoelectricity.
-
Since the authors are also experts in Hopfions, I would have liked a longer discussion on the differences and similarities between vortices and Hopfions and why the later are not observed here. The authors mention the Arnold theorem, but this is not clear to me. Why is the anisotropy the key parameter? Shouldn't it be the geometry?
-
p.8 "At larger radii, R>14nm..." this statement is valid for h = 6nm.
-
Looking at Figure 7, as mentioned before, one sees a drastic spatial change of the strain components. This means that the PbTiO3 unit cells are drastically deformed, especially near the vortex core. The calculations are based on the bulk parameters. How valid is this for unit cells that are so much deformed? It would be nice if the authors could comment on that.
-
Figure 9b, one observes a clear deviation from the linear scaling for large R - any comment/explanation?
-
Typo p.3: reach -> rich
Strengths
Weaknesses
Report
Requested changes
no further requests

---

## Round 2 · Author Response

We acknowledge the Editor and Referees for the careful reading and review of our work. We thank the Reviewers that they found our work interesting, recommended it for publication in SciPost Physics, and provided constructive comments for improving the manuscript. We have accounted for these insightful comments and addressed the Referees’ concerns point-by-point below.
Answers to the Report 1 on 2022-7-7
- Reviewer: Equation (5) gives the polarization distribution of an isotropic vortex in the cylindrical nanodot. However, this expression would not fulfill the natural boundary condition of zero polarization derivatives at r=R. Comment on this would be useful in the manuscript.
Answer: The solution given by Equation (5) is approximate and corresponds to the long-range asymptotic behaviour of polarization when r≫ξ0. Accordingly, the polarization derivative at r=R scales as (ξ0/R) and thus is approaching zero when ξ0/R ≪ 1. In this limit, it fits the natural boundary conditions and is in agreement with the numerical simulations as seen in Fig. 7a of the resubmitted manuscript. We thank the Reviewer for pointing out this issue and have included the corresponding remark in Appendix A.2, where the solution for the radial polarization distribution is obtained (page 14).
- Reviewer: In numerical simulations, the ferroelectric cylinders are surrounded by non-ferroelectric medium as shown in Fig.5. In the text, the cylinders are described as free-standing particles. I think it would be practical to mention the requirements for this surrounding media needed to consider the cylinders as free-standing, as well as to discuss the possible technological realizations of such free-standing nanostructures.
Answer: In our study, we consider the cylinder being embedded in vacuum with ε=1. For technical reasons of calculations, the stiffness coefficients were considered finite but very small that practically corresponds to the vacuum situation. Therefore our model corresponds to the free-standing nanostructures that are frequently realized in practice; see, for instance, Ref.24.
Answers to the Comment on 2022-05-18
- Reviewer: The authors stated that the basic idea of the polarization vortices formation is sketched in Fig. 1. Under short-circuited boundary conditions, the uniform polarization occurs in the PbTiO3 cylinder, while the vortex states are observed in the PbTiO3 cylinder under open-circuited boundary conditions. However, as reported by Li S. et al. (Appl. Phys. Lett. 111, 052901 (2017)), the symmetry of the electrical boundary conditions played more crucial influences on the formation of flux-closure domains. Thus, additional theoretical results or discussions are suggested to be included in this paper to illuminate the formation and transition of vortex states in ferroelectric cylinders.
Answer: We thank the Reviewer for drawing our attention to this research, which focuses on the important role of the electrodes in formation of ferroelectric domains. Indeed, contact with oxide electrodes may affect the depolarization field screening and lead to changes in the domain structure. However, in our model sketched in Fig.1 we consider ideal metallic electrodes with much smaller screening length compared to oxide electrodes discussed by Li S. et al. These considerations are beyond the scope of our work, which focuses on the vortex state formation in free-standing cylinders, but present an appealing problem for the future. Therefore, we add the corresponding remark to Discussion, Section 4.1, and highlight the mentioned research in References.
- Reviewer: Figure 7 is suggested to be included in the main text, which would facilitate to understand the transition between a-vortex states and c-vortex states displayed in the phase diagram in Fig. 2.
Answer: We have moved Figure 7 from Appendix A.3 to Section 4.4 of the main text as suggested (in the resubmitted version, it becomes Figure 4 on page 9).
- Reviewer: About the title, “in a ferroelectric cylinder” is suggested to be more specific, as in the manuscript, only PbTiO3 cylinder was discussed.
We have changed the title to “Vortex states in a ferroelectric PbTiO$_3$ cylinder” to be more specific as suggested by the Reviewer.

---

## Round 2 · List of Changes

1. The title was changed to “Vortex states in a ferroelectric PbTiO$_3$ cylinder”.
2. Section 4.1 was expanded with a discussion on possible types of electrodes for depolarization field screening.
3. Figure 7 was moved from Appendix A.3 to Section 4.4 of the manuscript (Fig 4 on page 9 in the resubmitted version).
4. A comment on polarization boundary conditions was added to Appendix A.2.
5. Reference [28] was added.

---

## Round 3 · Referee Report · Anonymous (Referee 2) · 2022-11-22

Strengths

Same as previous report (Anonymous Report 2 on 2022-9-19)

Weaknesses

None - the issues raised in the previous report have been fully addressed.

Report

The authors have answered all the questions raised in my previous report and made the appropriate modifications to the manuscript. I'm fully satisfied with their detailed answers and believe the paper is of high scientific quality. I highly recommend the publication of the paper in its present from.

Requested changes

None - all previously requested changes have been satisfyingly brought to the manuscript.

---

## Round 3 · Author Response

Dear Editor,

please find the updated version of our manuscript, with comments from Reviewer 2 taken into account.

We thank the Reviewer for careful reading of the manuscript and helpful suggestions. We have implemented the requested changes and corrected the typos. Below, we provide a detailed point-by-point reply to all the comments made.

Answers to the Report 2 on 2022-9-19

  1. Reviewer: In the case of a-vortices, how is the orientation of the in-plane vortex-axis determined? Is it artificially aligned with the [100] or [010] axis or can it take any other in-plane direction?

Answer: in our simulations, the system relaxes from the paraelectric state to $a$- (or $b$-) oriented vortex without any predefined vortex axis direction. If we detect several metastable states we select the one with the lowest energy. Therefore we are confident that we always obtain the stable state. In fact, the vortex core is aligned with either [100] or [010] axis due to the crystal symmetry, both $a$- and $b$-state being energetically equivalent. We added a note to Section 4.2.

  1. Reviewer: p.4 When stating that the c-state occurs only in very long cylinders with h>500nm: shouldn't it also vary as a function of R?

Answer: indeed, the resulting state depends on the aspect ratio and thus varies as a function of $R$; namely, the $c$-phase region becomes shorter as the radius increases. However the further investigation of bigger cylinders is challenging in terms of ‘valuable result’ to ‘power consumption spent on calculation’ relationship. We have commented on this aspect in Section 3.

  1. Reviewer: p.5 Figure 2 The solid blue curve visible on the phase diagram is explained only later (p.9, Fig.4 + text) - this information might be added already in the caption of Figure 2.

Answer: we thank the Reviewer for pointing out this issue. We have added the explanation in Section 3 and the caption of Fig.2 as suggested.

  1. Reviewer: p.6 The authors mention that "the observed perturbed polarization texture at cylinder edges possesses the spontaneously broken chirality, the effect that was thought to occur due to the flexoelectric contribution" - it would have been nice if the authors extended a bit this very interesting discussion. Indeed from Fig. 7, one sees that the strain gradients are very large, therefore one expects an important role played by flexoelectricity.

Answer: we have run a series of additional calculations with account of flexoelectric coupling term, $-\frac{1}{2}f_{ijkl}\left( P_{k}\partial_{l}u_{ij}-u_{ij}\partial {l}P\right) $, using the typical values of flexoelectric tensor components found in literature (Ponomareva et al, Phys. Rev. B 85, 104101(2012); Yudin and Tagantsev, Nanotechnology 24, 432001 (2013)). We don’t observe any significant impact of flexoelectricity on the resulting state, i.e. no new states in addition to those shown in the phase diagram (Fig.2). For instance, Fig.A1 (see the link attached to the List of changes) demonstrates the comparison between the polarization and strain distribution in Fig.7 of the manuscript and the distributions obtained with account of the flexoelectric term. Although the strain gradients are indeed considerable, the typical coupling coefficients are small, which limits the role of flexoelectricity.

  1. Reviewer: Since the authors are also experts in Hopfions, I would have liked a longer discussion on the differences and similarities between vortices and Hopfions and why the latter are not observed here. The authors mention the Arnold theorem, but this is not clear to me. Why is the anisotropy the key parameter? Shouldn't it be the geometry?

Answer: we are pleased to discuss the possibility of Hopfions emergence in ferroelectrics since, together with vortices, they form the building blocks of the topological ferroelectricity. In fact, the geometry, i.e. the geometrical confinement, enables the application of Arnold’s theorem to our system. According to it, the divergenceless vector field inside the restricted volume may swirl either in vortex or in Hopfion. Among these two, the more energetically favourable configuration is realised. In turn, the system energy is driven by the crystal anisotropy. As we already stressed in our previous publication, Ref.[30], the pronounced anisotropy of PTO favours the formation of vortices; however, in less anisotropic PZT the appearance of Hopfions is expected. Our current study on PTO cylinders confirms this statement: only more energetically favourable vortices are observed due to the relatively strong anisotropy. We expect, however, and this is confirmed by preliminary simulations, that Hopfions emerge in the similar geometry for PZT ferroelectrics. We have extended the discussion on geometry and anisotropy in Section 5.

  1. Reviewer: p.8 "At larger radii, R>14nm..." this statement is valid for h = 6nm.

Answer: we observe that the fragmentation of the system on multiple vortices occurs at radii R>14nm, at h=6-12 nm, this is region VII in diagram in Fig.2. We have corrected the statement correspondingly.

  1. Reviewer: Looking at Figure 7, as mentioned before, one sees a drastic spatial change of the strain components. This means that the PbTiO3 unit cells are drastically deformed, especially near the vortex core. The calculations are based on the bulk parameters. How valid is this for unit cells that are so much deformed? It would be nice if the authors could comment on that.

Answer: Indeed, within our approach the strain distribution close to the vortex core is challenging to calculate analytically, so we can reconstruct only the long-range asymptotic behaviour (15) as shown in Fig. 7b by dotted lines. This is a common methodology of the Ginzburg-Landau approach for calculations of vortices, for example in superconducting materials and superfluids, when the calculation of the vortex core energy is beyond the lowest-gradient-expansion approach and is treated as phenomenological parameter, using the extrapolation of the data from the low-gradient region. In our case the value of the unit cell deformation can be indeed underestimated, however, this does not affect the overall structure of the observed vortices at the large scale.

  1. Reviewer: Figure 9b, one observes a clear deviation from the linear scaling for large R - any comment/explanation?

Answer: the scattered points in Fig.9 correspond to the a-vortex states close to region II (twisted DW state) in diagram in Fig 2, where the vortex core may be already distorted and thus the assumption given by (4) can deviate. We have added the explanation in Section A.3.

  1. Reviewer: Typo p.3: reach -> rich

Answer: many thanks for careful reading. We have fixed the typo.

---

## Round 3 · List of Changes

1. A note considering the a-vortex axis orientation added to Section 4.2.
2. Section 3 is expanded with the discussion on the size of $c$-phase region in longer cylinders (nanowires) and the explanation of the separation line in Fig.2 (also in the caption to Fig.2).
3. A comment on Hopfions and the role of anisotropy is added to Section 5.
4. The explanation of scattered points in Fig.9 is added to Section A.3.
5. The statement at p.8 and a typo at p.3 are corrected.

All the changes are highlighted in red in the revised manuscript (see the link below).
https://drive.google.com/drive/folders/1vRXKjscBuaWrnKKU6YeBti6Tfbo2yho3

---

## Editorial Decision

published